# Predicting Early Death in Head and Neck Cancer—A Pilot Study

**DOI:** 10.3390/cancers17020302

**Published:** 2025-01-17

**Authors:** Charbél Talani, Hans Olsson, Karin Roberg, Emilia Wiechec, Alhadi Almangush, Antti A. Mäkitie, Lovisa Farnebo

**Affiliations:** 1Division of Sensory Organs and Communication, Department of Biomedical and Clinical Sciences, Linköping University, 581 83 Linköping, Sweden; lovisa.farnebo@regionostergotland.se; 2Region Östergötland Anaesthetics, Operations and Specialty Surgery Center, Department of Otorhinolaryngology, 582 25 Linköping, Sweden; karin.roberg@liu.se (K.R.);; 3Department of Pathology, Clinical and Experimental Medicine, Medical Faculty, Linköping University, 581 83 Linköping, Sweden; hans.olsson@regionostergotland.se; 4Division of Cell Biology, Department of Biomedical and Clinical Sciences, Linköping University, 581 83 Linköping, Sweden; 5Institute of Biomedicine, Pathology, University of Turku, 20014 Turku, Finland; 6Department of Pathology, University of Helsinki, 00014 Helsinki, Finland; 7Division of Ear, Nose and Throat Diseases, Department of Clinical Sciences, Intervention and Technology, Karolinska Institutet and Karolinska University Hospital, 171 76 Stockholm, Sweden; antti.makitie@helsinki.fi; 8Department of Otorhinolaryngology, Head and Neck Surgery, Helsinki University Hospital and University of Helsinki, 00029 Helsinki, Finland; 9Research Program in Systems Oncology, Faculty of Medicine, University of Helsinki, 00014 Helsinki, Finland

**Keywords:** head and neck squamous cell carcinoma, head and neck cancer, early death, early mortality, biomarkers, survivin

## Abstract

This study aimed to identify markers that might help predict the likelihood of early death in patients with head and neck cancer (HNC). Researchers examined tumor samples from Swedish HNC patients, focusing on those who died within six months of diagnosis and comparing them to patients who survived for at least two years. They found that patients who died early had higher levels of two markers—Ki-67 and survivin—compared to the survivors. Ki-67 was 21% higher in early-death patients, indicating more rapid cell growth. Survivin was also more intensely present in early-death patients’ cells. These findings suggest that measuring Ki-67 and survivin levels could help identify patients at a higher risk of early death, aiding in treatment planning.

## 1. Introduction

Early death in head and neck cancer (HNC) patients was previously defined as death within six months of diagnosis [1,2]. In Sweden, 9.5% of all patients with HNC died within this time. From 2008 to 2020, a decrease in early death for patients with a curative treatment decision was noted from 4.5% to 2.5% [3,4]. Age, WHO score, tumor site, advanced stage, and treatment modality were found to be independent factors for the prediction of early death [1]. Most HNC patients are diagnosed in the late stage of disease [5], and more rapid assessments of prognosis and easily accessible prognostic factors are needed to determine the best possible treatment recommendations.

Several potential predictive biomarkers have previously been explored in vitro and in vivo by our group and others [6,7,8,9]. In this study, a set of biomarkers previously shown to be of value for treatment sensitivity were selected to evaluate whether they were also of importance for the risk of early death.

The immunohistochemistry (IHC) staining of the expression and intensity of Ki67, survivin, WRAP53β, p16, EGFR, CA9, and fibronectin has previously been determined to be of prognostic and/or predictive value for HNC patients [7,8,9]. Other biological characteristics in the biopsy tissue such as tumor growth pattern, dissociation, i.e., tumor budding, stromal reactions like desmoplasia, the local immune response, and the tumor–stroma ratio could also contribute to predict prognosis for HNC patients [10,11].

Ki67 is a biomarker for proliferation, where a high percentage of positive cells reflects the tumor’s growing capacity [12]. High levels of Ki67 indicate rapid cell growth and invasiveness [13], making it a biomarker of interest when aiming to predict early death. Survivin (BIRC5) is a regulator of mitosis and inhibitor of apoptosis [14]. High nuclear survivin expression has been reported as a positive factor regarding disease-free survival in laryngeal cancer and advanced oral cavity cancer [7,15] and was therefore included in this study.

The scaffold protein WRAP53β governs the intracellular movement of DNA repair proteins, and its deficiency has been correlated with the onset of carcinogenesis [16]. Our earlier studies showed that positive WRAP53β in the cytoplasm combined with its negative expression in the cell nucleus was associated with poor outcome after radiotherapy for laryngeal glottic, oral cavity, and hypopharyngeal cancer [7].

P16 is a surrogate marker for Human Papilloma Virus (HPV) positivity. HPV-mediated oropharyngeal cancers are more sensitive to treatment and have better prognosis compared to HPV-negative tumors [17]. HPV has not been shown to have prognostic effects in cancers outside of the oropharyngeal region [18], but we aimed to evaluate if p16 expression and/or intensity could be correlated to early death, which, to our knowledge, has not been described previously.

The epidermal growth factor receptor (EGFR) is a cell surface receptor found mainly in cells of epithelial origin and present in up to 90% of head and neck squamous cell carcinoma (HNSCC) biopsies [19]. Mutations in the EGFR regulation can lead to oncogenic processes [20], and its overexpression is associated with poor prognosis in HNC [21].

Carbonic anhydrase-9 protein (CA9) is a glycoprotein involved in pH regulation [22], and it is a biomarker for tumor hypoxia [23]. The overexpression of CA9 is seen in different tumor types including breast, renal, and intracranial tumors, and is associated with poor prognosis [24]. In HNC, CA9 has been associated with reduced overall and disease-free survival [25].

Fibronectin is a secretory protein that enhances the permeability of endothelial cells in vascular walls by binding to receptors [26]. Its overexpression has been shown to be essential for HNC migration and invasion [27] and has been correlated with a higher stage of disease and mortality [28]. Thus, it was included in this study.

Furthermore, the biological characteristics of tumor growth patterns, such as immunological response in the tumor tissue and tumor budding, were evaluated since they have been shown to predict tumor aggressiveness in HNC patients [29,30,31]. The lymphocytic host response and tumor-infiltrating lymphocyte (TIL) count are immunological responses to tumor growth and were first described in breast cancer [32] but have also been found to have an adverse prognostic value in HNC [33]. The tumor stoma ratio (TSR), which describes the proportion of tumor tissue relative to the surrounding stroma, has been described as a prognostic marker but has not been evaluated for early death [34]. Tumor budding is characterized by a single cell or a group of cells in a cluster in the invasive front of the biopsy, and it is an adverse prognostic factor in HNC, although it has not been correlated with early death [10].

In this pilot study, our aim was to combine the IHC staining of biomarkers with biological characteristics detected in hematoxylin–eosin specimens of biopsy tissue to evaluate the risk of early death. Furthermore, we wanted to investigate whether any combination of markers could be used as a prognostic aid for the prediction of death within six months.

## 2. Materials and Methods

From medical records, we retrospectively identified 43 head and neck squamous cell carcinoma (HNSCC) patients with curative treatment intent who died within six months of diagnosis between 2008 and 2015. During the same period, a dataset containing biopsies from HNSCC patients in which tumor material could be extracted for research purposes without affecting the histopathological diagnosis was collected (biobank No 416, the National Board of Health and Welfare in Sweden) at the Department of Otorhinolaryngology, Head and Neck Surgery, the University Hospital of Linköping, Sweden (approved by the Ethical Committee of Stockholm 2017/1035-31/2). Twelve biopsies from patients who died within six months, according to the medical record review, were found in the biobank. Three biopsies were excluded: two being too small for an additional investigation and one biopsy without malignant areas on the IHC glasses. After exclusion, 9 early-death patients were identified.

From the same tumor collection, as many matching survivors as possible according to tumor site and stage were selected (*n* = 17). Patients in the survivor group all had a survival time greater than two years after diagnosis (Table 1). A total of 26 HNSCC biopsies with curative treatment between 2008 and 2015 were selected for this pilot study. For all samples, follow-up data on clinical parameters of the patients were available from the Department of Otorhinolaryngology, Head and Neck Surgery, Linköping University Hospital, Sweden.

### 2.1. Immunohistochemistry

The immunohistochemical staining of the HNSCC sections was performed as previously described [35]. Briefly, 5 µm thick tumor sections were dried, deparaffinized in Histolab-Clear (Histolab, Gothenburg, Sweden) and rehydrated in alcohol. Antigen retrieval (PT 200, DAKO Denmark A/S, Glostrup, Denmark) was performed, and the sections were blocked for endogenous peroxidase activity in 3% H_2_O_2_ (Sigma–Aldrich, St. Louis, MO, USA) and thereafter incubated in blocking buffer (0.1% BSA-5% FBS in TBS-Triton). The sections were incubated at 4 °C overnight with primary antibodies against Ki67 (1:50; Santa Cruz Biotechnology, Dallas, TX, USA), EGFR (1:50, Cell Signaling Technology, Danvers, MA, USA), CA9 (1:400; Novus Biologicals, Littleton, CO, USA), WRAP53ß (1:1000; Innovagen AB, Sweden), survivin (1:100; Thermo Fisher Scientific, Waltham, MA, USA), fibronectin (1:200; Sigma–Aldrich, St. Louis, MO, USA), and p16 from Roche Ventana (Tucson, AZ, USA), and IHC was performed using antibodies against CINtec R p16INK4a and clone E6H4 according to the manufacturer’s protocol.

After washing in TBS/0.3% Triton X-100, the sections were incubated with a goat anti-rabbit or a goat anti-mouse HRP-conjugated antibody (1:500; IgG, EMD Millipore Corporation, Temecula, CA, USA) and incubated with an ImmPACT NovaRED peroxidase substrate kit (Vector Laboratories, Burlingame, CA, USA). Finally, the sections were counterstained with Mayer’s hematoxylin (HistoLab), dehydrated, cleared in Histolab-Clear, and mounted with Pertex (HistoLab).

As negative controls, sections were stained as described above but without the primary antibody. Images were acquired with a light microscope (Olympus BX51, Shinjuku City, Tokyo, Japan) with a ×20 objective.

The IHC scoring of sections stained with EGFR, fibronectin, survivin, CA9, p16, and WRAP53β was performed by one pathologist (HO) and one head and neck surgeon (CT) who were blinded to patient outcome and treatment response. For slides with differing scores, a definite consensus score was determined by mutual agreement in a separate session. For WRAP53β, survivin, CA9, fibronectin and EGFR, both the percentage of positively stained cells **0:** 0%, **1:** <10%, **2:** 10–50% or **3:** >50% and the intensity of staining as follows: **0:** none, **1:** weak, **2:** moderate or **3:** strong were evaluated. For WRAP53β and survivin, staining was predominantly nuclear. The intensity of staining in the cell nucleus and cytoplasm was scored as follows: **0:** none, **1:** weak, **2:** moderate or **3:** strong. Intensity and percentage scores of 2 or 3 were considered positive. Ki67 was estimated as a percentage ratio: (number of stained nuclei divided by number of all nuclei) × 100%. In the analysis, hotspot areas were selected, defined as areas with the highest number of positive tumor nuclei [36]. For p16, both cytoplasmic and nuclear expression and intensity were estimated as follows: 0: no positive cells; 1: 1–25% cells positive; 2: 26–50% cells positive; and 3: 51–100% cells positive. A staining rate above 70% was considered positive in accordance with the College of American Pathologists Guidelines [37].

Histopathological markers in HE-stained sections were assessed as described in recently published guidelines [38,39,40,41]. The infiltration of TILs was defined as the percentage of tumor stroma infiltrated with lymphocytes, <20% was defined as low infiltration, and ≥20% as high infiltrating lymphocytes in the biopsies. The TSR was categorized as low stroma (<50%) or high stroma (>50%) occurrence. Two categories of tumor budding were defined: <5 buds was considered low-intensity budding, while ≥5 buds was considered high-intensity budding.

### 2.2. Statistics

The data are presented as the mean, standard deviation, and range for continuous variables and as numbers and percentages for categorical variables. For comparisons between two groups, the independent *t*-test and the Mann–Whitney U-test was used. For the variables Ki-67, TIL, and tumor budding, equal variances between the groups were not assumed. All other variables were categorical, and the chi-square test was used for comparisons between them, and Fisher’s exact test was used for comparisons between dichotomous variables. Exact binominal confidence intervals were estimated for proportions. A Kaplan–Meier plot was used to describe the survival for the subgroups, and the difference between subgroups was analyzed with a log-rank test. Cox regression was used for multivariable analyses. All significance tests were two-tailed and were conducted at the 5% significance level. IBM SPSS Statistics for Macintosh, Version 29.0. (IBM Corp., Armonk, NY, USA), was used for all statistical analyses.

## 3. Results

### 3.1. Patients

IHC staining was used to detect differences between two groups of patients, “early death” dying within six months of diagnosis (*n* = 9) and matched “survivors” who lived for at least two years after diagnosis (*n* = 17) (Table 1).

Of the nine early-death patients, three had oral cavity cancer, three had laryngeal cancer, two had oropharyngeal cancer, and one had hypopharyngeal cancer. Five patients were female, and four were male (Table 1). A total of 17 patients were matched survivors: seven patients with oral cavity cancer, five with laryngeal cancer, four with oropharyngeal cancer, and one with hypopharyngeal cancer. Seven patients were female and ten male (Table 1).

### 3.2. Biomarkers

The expression and intensity of markers were compared between the early-death patients and survivors. There was a significantly higher percentage of Ki-67-positive cells in patients who died within six months compared to those surviving for two years, and the mean difference was 21% (65.7% vs. 44.6% CI 95% for difference 1.3–40.7) (*p* = 0.038) (Figure 1 and Figure 2; Table 2).

We found a significant difference in cytoplasmic survivin expression, where early-death patients had increased expression compared to survivors (*p* = 0.024). The positive survivin staining was correlated with early death in the analyzed HNSCC patients (Figure 3 and Table 3). Furthermore, the staining intensity for survivin differed between the groups (*p* = 0.006) (Figure 2 and Figure 4). In the early-death group, 89% had survivin intensity grades 1 and 2 (Figure 4) compared to 41% in the survivor group. In total, 11% of the early-death patients had no survivin staining in their biopsy tissue compared to 59% of the survivors (*p* = 0.019).

For the other investigated biomarkers, no significant differences in mean expression or intensity were found between the groups (Table 2). There was a trend towards higher p16 expression positivity in the surviving group, but no difference in intensity was noted between the groups. EGFR and CA9 both had a tendency towards higher expression and intensity in the early-death group. For WRAP53β and fibronectin, there were no differences between the groups (Table 2).

Cox regression was performed, including Ki-67 and survivin adjusted for site and stage. Survivin was an independent adverse prognostic factor for early death (HR 3.5 [CI 95% 1.02–11.8] *p* = 0.041); however, Ki-67 was not an independent prognostic factor for early death in the multivariable analysis.

No significant differences were found between the groups regarding TILs (*p* = 0.259), TSR (*p* = 0.142), or tumor budding (*p* = 0.504).

The low infiltration of TILs was observed in 56% of all early-death patients and 63% of the surviving patients. On average, TIL expression was 4.8% lower in the early-death group than in the survival group (19.6% vs. 24.8%). A low TSR was found in 78% of all early-death patients compared to 56% of the patients in the surviving group, and low tumor budding was found in 78% and 75%, respectively.

## 4. Discussion

To the best of our knowledge, this pilot study is the first to examine whether a set of biomarkers present in the tumor at the time of diagnosis will have prognostic value for early death in HNC patients. We chose to investigate a panel of biological markers that were previously shown to be of interest in HNC to determine whether these markers could be used to predict early death in HNC patients.

In this study of nine patients with early death and 17 matched survivors, we found a significant difference in the expression of Ki-67 and survivin between patients dying within six months and those surviving for at least two years. Our findings suggest that information from biopsy material could be used in prediction for early death among HNC patients and possibly play a role in treatment decision making.

The treatment of choice for HNC is surgery and/or (chemo)radiotherapy. Treatment regimens are known for their many side effects and effect on quality of life. Thus, it is of the utmost importance to identify patients at risk of dying within six months to provide the patient with an opportunity to evaluate his/her treatment options. If a high risk of early death is identified, treatment intensity discussions would be valuable. An analysis of biomarkers and histological features could play a greater role at multidisciplinary tumor board meetings in the future, given that a reliable panel was identified [42]. If panels of prognostic markers were identified in a greater quantity of material, discussions on the de-escalation of treatment could have a place for patients with a short, estimated survival time in order to maximize the quality of the remaining life for patients and to avoid hospitalization and complications after intense treatment regimens.

The cohort in this study was small because even if HNC is a potentially lethal disease, it is still rare for patients to die within six months (only approximately 9–11%) [1,2]. Due to the deadly nature of hypopharyngeal carcinoma (only 25% 5-year overall survival) [43], it was not possible to include more than one patient who survived for 24 months, which also matched the stage of the early-death patient.

We showed that a high percentage of Ki-67 cells was correlated with a poor prognosis, in line with other studies regarding Ki-67 as a marker of poor prognosis [12,44]. In contrast, Maebayashi et al. showed that a high expression of Ki-67 in combination with p16 positivity was a prognostically beneficial marker for HNC of unknown primary (CUP) in a small study of 13 patients [45], which contradicts our results. However, no CUPs were included in our material.

The influence of survivin on prognosis has been described and debated in various studies because of the divergent results. We have shown that nuclear survivin expression is correlated with a good response to radiotherapy and is a positive predictive factor for treatment response [8]. Others have shown shorter overall survival, and a high tumor stage is correlated with high cytological survivin expression [46]. High expression was correlated with more aggressive disease in breast, hematological, colorectal, gastric, and urine bladder cancer [47], in line with the results of this study, suggesting that cytoplasmic survivin is also an indicator of poor prognosis in HNC patients.

Trends regarding p16, CA9, and EGFR were noted; however, no significant differences between the groups were observed. p16, routinely used as a surrogate marker for HPV positivity, was analyzed for both expression and intensity in this study since tumors from various sites and not only from the oropharynx were included. A future analysis of a greater quantity of material of HNC patients in relation to p16 intensity could be of interest. High EGFR expression is well known for its correlation with poor prognosis [48]. Thus, we expected a correlation between high expression and intensity in the early-death group but could not verify this hypothesis, possibly due to the small number of included patients. CA9-overexpressing tumors are more resistant to cisplatin, as shown in previous studies [49]. In our pilot study, there was a non-significant tendency toward a higher and more intense expression of CA9 in the early-death group, suggesting that an analysis of a larger cohort would be of interest.

The high expression of TILs has been described as a beneficial prognostic factor [50]; in this small study, we observed a positive tendency, although the difference was not significant. Larger studies are warranted to analyze the importance of TILs for early death. Our belief was that there would be higher tumor budding and a greater TSR in the early-death group, in line with previous studies showing that high budding and a greater TSR correlated to worse prognosis [10,34]. However, tumor budding and the TSR were not significantly different between the two cohorts.

Further studies analyzing factors of importance for early death could include liquid biopsies, particularly circulating tumor-DNA based assays (ctDNAs), which are revolutionizing cancer care by enabling personalized diagnostics and surveillance cost-effectively [51,52] and are easily accessible [53]. CtDNA has shown promising prognostic value for HNSCC [54] and correlates to recurrence in both HPV-associated oropharyngeal cancer [55] and non HPV-related disease [56,57,58]. While protocols are still evolving, ongoing research and clinical trials pave the way for these technologies to become standard practice in managing head and neck cancers and could be included in future studies predicting early death.

Limitations to this study were the small sample size of 9 early-death patients and 17 matched survivors. The study design was a retrospective analysis of an existing sample collection for a panel of markers to analyze its usefulness for the prognostication of sur-vival. This study highlights potential predictive markers for survival, but further validation in larger, retrospective or prospective studies is needed to safely develop a panel suitable for clinical use.

## 5. Conclusions

Taken together, our results indicate that survivin and Ki67 can be suggested as potential prognostic biomarkers for early death and could be included in a panel of markers predicting early death in patients with HNSCC. If verified in more material, these factors can assist in treatment decision making in the future.

## Figures and Tables

**Figure 1 cancers-17-00302-f001:**
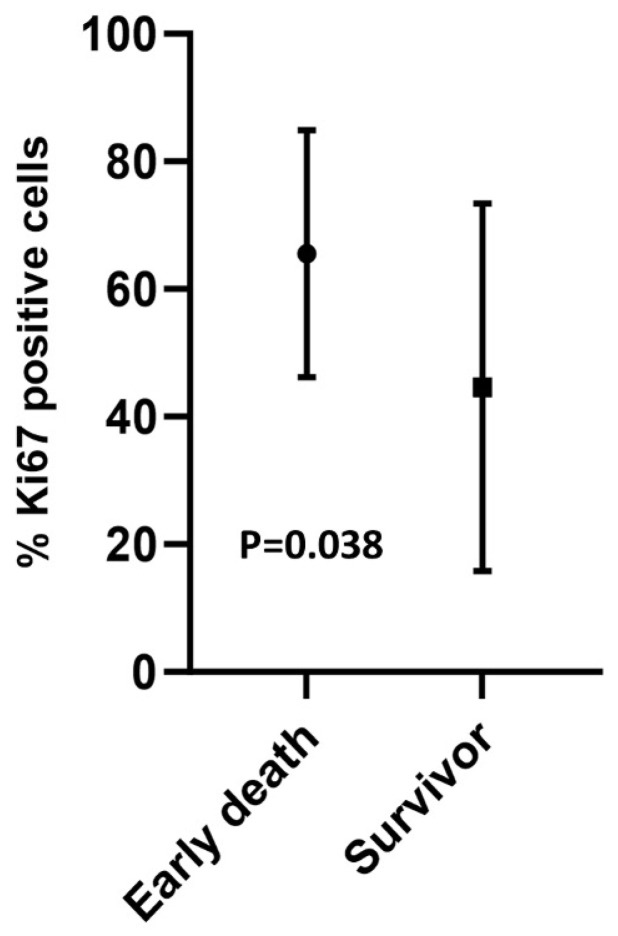
Expression of Ki67 in 9 early-death patients and 17 survivors (95% confidence interval); the dot and square indicate the mean Ki67 expression. *p*-value analyzed using Mann–Whitney U-test.

**Figure 2 cancers-17-00302-f002:**
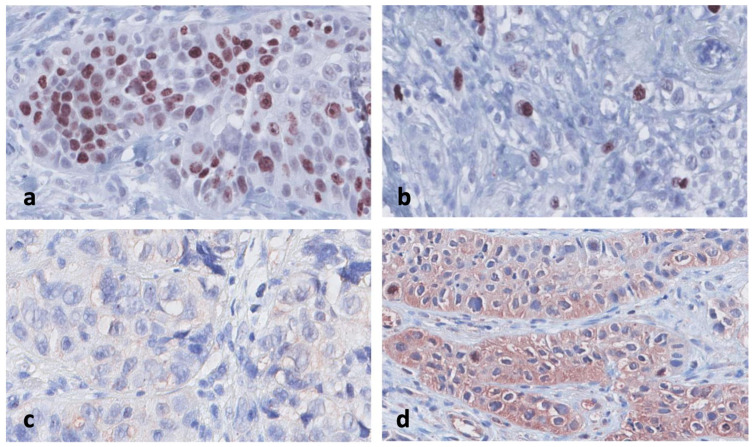
(**a**) Ki67 with hotspot count greater than 80%. (**b**) Ki67 with hotspot count less than 20%. (**c**) Survivin with no intensity in the cytoplasm. (**d**) Survivin with intensity grade 2 in the cytoplasm. Ki67 was estimated as a percentage ratio: (number of stained nuclei divided by number of all nuclei) × 100%. In the analysis, hotspot areas were selected, defined as areas with the highest number of positive tumor nuclei. For surviving, the percentage of positively stained cells was translated to a score [0 (0%), 1 (<10%), 2 (10–50%) or 3 (>50%)]. The intensity of staining was scored as follows: [0 (none), 1 (weak), 2 (moderate) or 3 (strong)].

**Figure 3 cancers-17-00302-f003:**
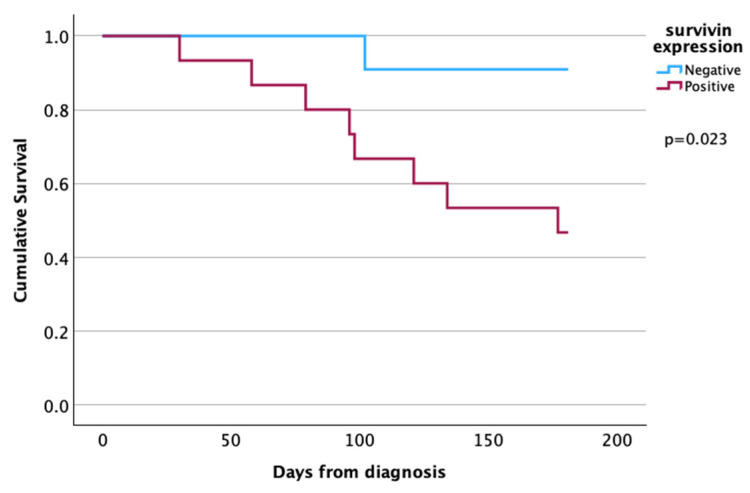
Kaplan–Meier curve of death within six months based on expression of survivin. *p*-value analyzed using log-rank.

**Figure 4 cancers-17-00302-f004:**
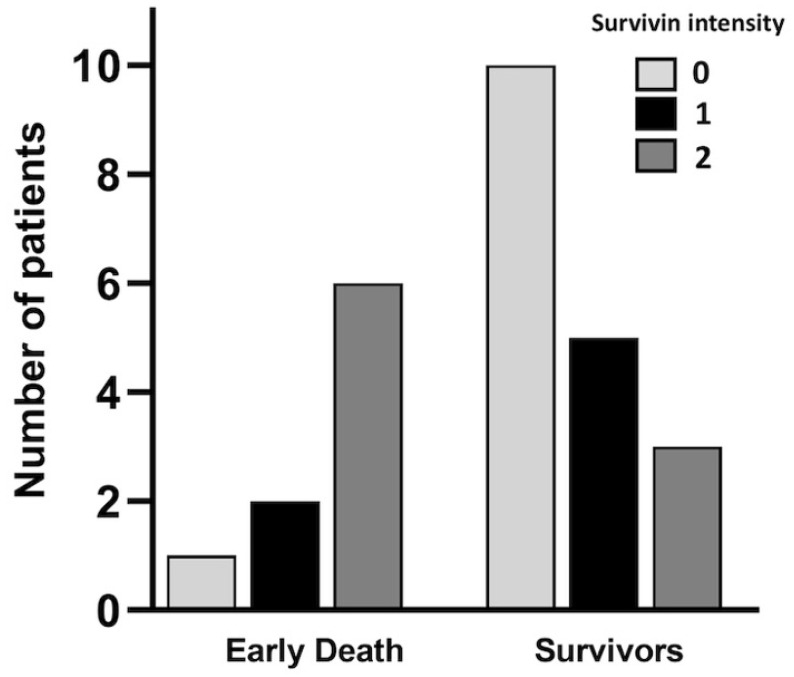
Survivin intensity in early-death patients and survivors.

**Table 1 cancers-17-00302-t001:** Descriptive data showing cases (*n* = 9), 1–9 being early-death patients, and controls (*n* = 17) being matched to survivors, colors represent different tumor sites. # = number, CRT = Chemoradiotherapy, RT = Radiotherapy.

Site	Patient #	Stage	TNM 7	Survival Time	Smoking	Treatment	Sex
Oral Cavity	1	IV	T3N2bM0	4 m	Yes	CRT + Surgery	Female
	Control 1.1	IV	T3N2bM0	>24 m	Yes	RT + Surgery	Female
	Control 1.2	IV	T4aN0M0	>24 m	Previous	CRT	Male
	Control 1.3	IV	T4aN0M0	>24 m	Unknown	RT + Surgery	Female
Oral Cavity	2	III	T2N1M0	3 m	Yes	CRT + Surgery	Male
	Control 2.1	III	T2N1M0	>24 m	Yes	RT + Surgery	Male
	Control 2.2	III	T3N1M0	>24 m	Yes	RT + Surgery	Female
Oral Cavity	3	III	T3N1M0	5 m	Yes	RT + Surgery	Male
	Control 3.1	III	T2N1M0	>24 m	Unknown	RT	Male
	Control 3.2	III	T2N1M0	>24 m	Yes	RT	Female
Larynx	4	IV	T4N2cM0	4 m	Yes	CRT	Male
	Control 4.1	IV	T4aN1M0	>24 m	Yes	Surgery + RT	Male
	Control 4.2	IV	T4aN1M0	>24 m	No	Surgery + RT	Male
	Control 4.3	IV	T4aN1M0	>24 m	No	Surgery + RT	Male
Larynx	5	IV	T4aN1M0	5 m	Yes	No	Female
	Control 5.1	IV	T4aN1M0	>24 m	Yes	RT + Surgery	Male
Larynx	6	IV	T4aN2cM0	1 m	Yes	RT	Male
	Control 6.1	IV	T4aN1M0	>24 m	Yes	Surgery + RT	Male
Oropharynx	7	III	T2N1M0	4 m	Yes	RT	Female
	Control 7.1	III	T2N1M0	>24 m	Unknown	CRT	Male
	Control 7.2	III	T2N1M0	>24 m	Yes	CRT	Male
Oropharynx	8	IV	T3N2bM0	5 m	Previous	CRT	Female
	Control 8.1	IV	T4N2cM0	>24 m	Unknown	CRT	Female
	Control 8.2	III	T3N1M0	>24 m	No	RT	Female
Hypopharynx	9	IV	T4N2bM0	4 m	Yes	RT	Female
	Control 9.1	IV	T3N2M0	>24 m	Previous	RT	Female

**Table 2 cancers-17-00302-t002:** Comparison of the intensity and expression of markers between early-death patients and survivors using the chi^2^ test * or Student’s *t* test **.

Marker	Early Death	Survivor	*p*-Value
Survivin intensity *n* (%)			0.006 *
0	1 (11)	10 (59)	
1	2 (22)	4 (24)	
2	6 (67)	3 (17)	
3	0 (0)	0 (0)	
Ki-67 mean (SD)	65.6 (19.3)	44.6 (28.8)	0.038 **
p16 intensity *n* (%)			0.077 *
0	5 (56)	11 (65)	
1	1 (11)	0 (0)	
2	0 (0)	2 (12)	
3	3 (33)	4 (23)	
EGFR intensity *n* (%)			0.149 *
0	0 (0)	4 (24)	
1	1 (12)	4 (24)	
2	4 (44)	2 (12)	
3	4 (44)	7 (40)	
CA 9 intensity *n* (%)			0.089 *
0	4 (44)	9 (53)	
1	0 (0)	3 (18)	
2	4 (44)	4 (24)	
3	1 (12)	1 (5)	
WRAP53β intensity *n* (%)			0.133 *
0	5 (56)	12 (71)	
1	0 (0)	2 (12)	
2	4 (44)	3 (17)	
3	0 (0)	0 (0)	
Fibronectin intensity *n* (%)			0.203 *
0	3 (33)	3 (18)	
1	1 (11)	3 (18)	
2	2 (23)	3 (18)	
3	3 (33)	8 (46)	
TIL mean (SD)	19.6 (16.3)	24.3 (19.6)	0.259 **
TSR stromal intensity *n* (%)			0.264 *
Low	7 (78)	9 (56)	
High	2 (22)	7 (44)	
Budding expression *n* (%)			0.637 *
Low	7 (78)	12 (75)	
High	2 (22)	4 (25)	

**Table 3 cancers-17-00302-t003:** Comparison between expression of survivin between early-death patients and survivors. *p* = 0.019 using Chi^2^ test.

Survivin Expression *n* (%)	Early Death	Survivor
Positive	8 (89)	7 (41)
Negative	1 (11)	10 (59)

## Data Availability

The data presented in this study are available on request from the corresponding author due to the confidentiality of the study subjects.

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
