# Peer review of "Predicting Early Death in Head and Neck Cancer—A Pilot Study"

_cancers, 2025, doi:10.3390/cancers17020302_

Round 1

Reviewer 1 Report

Comments and Suggestions for Authors

The authors have done an interesting work to identify markers help predict early death in patients with head and neck cancer. However there are some correction which needs to be done before the article could be accepted.

1. The arounds should mention the number of patient cohort (n=?) studied in the legend for Table 1.

2. The statistical tool used to analyze the p value for % of Ki67 positive cells needs to be included in description of the Figure 1 legend.

3. Please insert a scale bar in Fig 2. Tissue histopathology image.

4. Please perform a statistical analysis of survival in Kaplan Meier plot (Fig 3).

5. Are there no replicates in Survivin intensity in early death patients and survivors? As there is no error bar in fig 4.

6. The authors should include some recent studies of detecting head and neck cancer using various methods such as ctDNA or cfDNA in their introduction or dicsussion (eg. https://pubs.acs.org/doi/10.1021/acsbiomaterials.4c00606; https://link.springer.com/article/10.1007/s10555-017-9685-x).

Author Response

On behalf of all the authors, I hereby submit our revised manuscript, Predicting early death in Head and Neck cancer – a pilot study by Charbél Talani, MD PhD; Hans Olsson, MD PhD; Karin Roberg, Professor; Emilia Wiehec, Associate Professor; Alhadi Almangush, Professor; Antti Mäkitie, MD, Professor; Lovisa Farnebo, MD, Associate Professor.

We would like to thank the reviewers for their insightful and constructive comments on the manuscript which we have carefully considered and revised the manuscript accordingly. All changes have been highlighted in yellow in the manuscript and the reviewers have received point-by-point responses to each comment with specific notations to manuscript changes and pages. We hope this revised manuscript is acceptable for publication in Cancers.

Sincerely,

Charbél Talani, corresponding author

Department of Otorhinolaryngology and Head and Neck surgery, Linköping, Sweden.

Response to the reviewer 1

The authors have done an interesting work to identify markers help predict early death in patients with head and neck cancer. However, there are some correction which needs to be done

Thank you for your relevant comments. We will address each issue below highlighted in yellow.

  1. The arounds should mention the number of patient cohort (n=?) studied in the legend for Table 1.

Thank you for this valuable comment. The table legend for Table 1 has been updated accordingly. See below – changes indicated in yellow.

Table 1. Descriptive data showing cases (n=9), 1-9 being early death patients, and controls (n=17) being matched survivors.

  1. The statistical tool used to analyze the p value for % of Ki67 positive cells needs to be included in description of the Figure 1 legend.

Thank you, the statistical tool used to analyze the p-value is now included in the figure legend.

P-value analyzed using Mann-Whitney U-test.

  1. Please insert a scale bar in Fig 2. Tissue histopathology image.

Thank you, information regarding the scales used for intensity has now been added to the figure legend.

Ki67 was estimated as a percentage ratio: (number of stained nuclei divided by number of all nuclei × 100%). In the analysis, hot spot areas were selected, defined as areas with the highest number of positive tumor nuclei. For survivin the percentage of positively stained cells was translated to a score (0 (0%), 1 (<10%), 2 (10–50%) or 3 (>50%)). The intensity of staining was scored as follows; (0 (none), 1 (weak), 2 (moderate) or 3 (strong)).

  1. Please perform a statistical analysis of survival in Kaplan Meier plot (Fig 3).

A statistical analysis using Log-rank test has now been added to the figure legend.

Figure 3. Kaplan-Meier curve of death within six months based on expression of survivin. P-value analyzed using Log-rank.

  1. Are there no replicates in Survivin intensity in early death patients and survivors? As there is no error bar in fig 4.

We had access only to a limited number of IHC-glasses per tumor in this study, hence the pilot study. That is why replications were not possible in this material. In the following study including 70 head and neck cancer cases, three replications of each antibody will be performed. However, this was unfortunately not possible in the current pilot study.

  1. The authors should include some recent studies of detecting head and neck cancer using various methods such as ctDNA or cfDNA in their introduction or dicsussion (eg. https://pubs.acs.org/doi/10.1021/acsbiomaterials.4c00606; https://link.springer.com/article/10.1007/s10555-017-9685-x).

A paragraph has been added to the discussion with new references regarding circulating tumor-DNA.

Further studies analyzing factors of importance for early death could include liquid biopsies, particularly circulating tumor-DNA based assays, (ctDNA) which are revolutionizing cancer care by enabling personalized diagnostics and surveillance cost-effectively [59, 60], and easily accessible [61]. CtDNA has shown promising prognostic value for HNSCC [62], and correlates to recurrence in both HPV-associated oropharyngeal cancer [63], and non HPV-related disease [64-66]. While protocols are still evolving, ongoing research and clinical trials pave the way for these technologies to become standard practice in managing head and neck cancers, and could be included in future studies predicting early death.

Thank you for all comments that helped to improve the manuscript.

Reviewer 2 Report

Comments and Suggestions for Authors

The topic discussed in this paper is of considerable interest nowadays, with a view to personalising treatment for head and neck cancer patients. Despite the small sample size, the study is well designed and reproducible, and lays the groundwork for reviewing the results on a larger sample size. That is why I am in favour of publication. 

Author Response

On behalf of all the authors, I hereby submit our revised manuscript, Predicting early death in Head and Neck cancer – a pilot study by Charbél Talani, MD PhD; Hans Olsson, MD PhD; Karin Roberg, Professor; Emilia Wiehec, Associate Professor; Alhadi Almangush, Professor; Antti Mäkitie, MD, Professor; Lovisa Farnebo, MD, Associate Professor.

We would like to thank the reviewers for their insightful and constructive comments on the manuscript which we have carefully considered and revised the manuscript accordingly. All changes have been highlighted in yellow in the manuscript and the reviewers have received point-by-point responses to each comment with specific notations to manuscript changes and pages. We hope this revised manuscript is acceptable for publication in Cancers.

Sincerely,

Charbél Talani, corresponding author

Department of Otorhinolaryngology and Head and Neck surgery, Linköping, Sweden.

Reviewer 2: The topic discussed in this paper is of considerable interest nowadays, with a view to personalising treatment for head and neck cancer patients. Despite the small sample size, the study is well designed and reproducible and lays the groundwork for reviewing the results on a larger sample size. That is why I am in favour of publication. 

Thank you for this comment.

Reviewer 3 Report

Comments and Suggestions for Authors

This manuscript concerns the study of a biomarker panel (mostly from immunohistochemistry) in relation to early death (<6month survival) in head and neck cancer patients. The authors concluded that cytoplasmic survivin and Ki-67 were correlated to early death. 

My main concern is that the above markers have not been correlated to important factors of survival and early death such as general health status and treatment complications. Thus, their conclusion that these markers may affect treatment decision making is at least arbitrary and cannot be supported by the results. Nevertheless, it is not useless and to my opinion the authors should mention this concern.

Other minor comments.

The author already report that they have a small sample. The fact that their sample includes a variety of  head and neck cancers eg from hypopharynx, larynx etc, further weakens their results

Cox regression analysis that they have performed is not shown. In contrast figures 1 and 4 are maybe useless.   

Author Response

On behalf of all the authors, I hereby submit our revised manuscript, Predicting early death in Head and Neck cancer – a pilot study by Charbél Talani, MD PhD; Hans Olsson, MD PhD; Karin Roberg, Professor; Emilia Wiehec, Associate Professor; Alhadi Almangush, Professor; Antti Mäkitie, MD, Professor; Lovisa Farnebo, MD, Associate Professor.

We would like to thank the reviewers for their insightful and constructive comments on the manuscript which we have carefully considered and revised the manuscript accordingly. All changes have been highlighted in yellow in the manuscript and the reviewers have received point-by-point responses to each comment with specific notations to manuscript changes and pages. We hope this revised manuscript is acceptable for publication in Cancers.

Sincerely,

Charbél Talani, corresponding author

Department of Otorhinolaryngology and Head and Neck surgery, Linköping, Sweden.

Reviewer 3: This manuscript concerns the study of a biomarker panel (mostly from immunohistochemistry) in relation to early death (<6month survival) in head and neck cancer patients. The authors concluded that cytoplasmic survivin and Ki-67 were correlated to early death. 

My main concern is that the above markers have not been correlated to important factors of survival and early death such as general health status and treatment complications. Thus, their conclusion that these markers may affect treatment decision making is at least arbitrary and cannot be supported by the results. Nevertheless, it is not useless and to my opinion the authors should mention this concern.

Thank you for your valuable comment.

It is correct that health status and treatment complications were not included in this study. This is a pilot study, simply comparing the expressions and intensities of a set of factors between survivors and non-survivors of HNC.

When conducting a larger study, including 70 HNC cases, to validate the results from the pilot study, WHO function class and treatment outcome will be included.

Other minor comments.

The author already report that they have a small sample. The fact that their sample includes a variety of head and neck cancers eg from hypopharynx, larynx etc, further weakens their results

This is a valid comment, thank you. Head and neck cancers are a heterogenous group and as this pilot study focused on the differences between two cohorts in a small sample size, all different sites available in the biobank were included. With a larger sample size we hope to confirm the results from this pilot study and if sample sizes allow possibly make subgroup analysis for the different sub-sites.

Cox regression analysis that they have performed is not shown. In contrast figures 1 and 4 are maybe useless.   

The cox regression is shown below, although not included in the manuscript. If you wish to include it please do so. 95% confidence intervals for Odds Ratio is included for the factors in the analysis.

Cox regression

   Sig.

                   OR

95,0% CI for OR

               Lower

          Upper

Survivin

.041

3.420

1.049

11.154

Ki67

.181

1.012

.956

1.071

Oral Cavity

.946

Oropharynx

.868

1.216

.122

12.130

Larynx

.704

.638

.063

6.477

Hypopharynx

.990

.982

.058

16.524

Stage

.426

2.684

.236

30.592